

# Design of an enhanced feature point matching algorithm utilizing 3D laser scanning technology for sculpture design

Xiaoxiong Zheng[1] and Zhenwei Weng[2]

[1] School of Sculpture and Public Art, Guangzhou Academy of Fine Arts, Guangzhou, Guangdong, China
[2] Creation and Arts College, Universiti Teknologi MARA (Uitm), Shah Anam, Selangor, Malaysia

## ABSTRACT

As the aesthetic appreciation for art continues to grow, there is an increased demand for precision and detailed control in sculptural works. The advent of 3D laser scanning technology introduces transformative new tools and methodologies for refining correction systems in sculpture design. This article proposes a feature point matching algorithm based on fragment measurement and the iterative closest point (ICP) methodology, leveraging 3D laser scanning technology, namely Fragment Measurement Iterative Closest Point Feature Point Matching (FM-ICP-FPM). The FM-ICP-FPM approach uses the overlapping area of the two sculpture perspectives as a reference for attaching feature points. It employs the 3D measurement system to capture physical point cloud data from the two surfaces to enable the initial alignment of feature points. Feature vectors are generated by segmenting the region around the feature points and computing the intra-block gradient histogram. Subsequently, distance threshold conditions are set based on the constructed feature vectors and the preliminary feature point matches established during the coarse alignment to achieve precise feature point matching. Experimental results demonstrate the exceptional performance of the FM-ICP-FPM algorithm, achieving a sampling interval of 200. The correct matching rate reaches an impressive 100%, while the mean translation error (MTE) is a mere 154 mm, and the mean rotation angle error (MRAE) is 0.065 degrees. The indicator represents the degree of deviation in translation and rotation of the registered model, respectively. These low error values demonstrate that the FM-ICP-FPM algorithm excels in registration accuracy and can generate highly consistent three-dimensional models.

# INTRODUCTION

Within China's urban renewal context, there is an increased focus on revitalizing and planning landscapes while also considering contemporary sculpture creation concepts, evaluations, and recognition of national construction development. As an integral element of landscape architecture, sculpture embodies traditional artistry alongside practicality and design innovation. Landscape space is central to forming aesthetic landscapes; a thoughtful integration of sculpture within these spaces can amplify the sculpture's value and enhance

Corresponding author
Zhenwei Weng,
wengzhenwei@xmedus.cn

the landscape's multifaceted character (*He et al., 2022*). Traditionally, sculpture design has relied on hand-drawn sketches or photographs, which often struggle to accurately represent intricate forms and details, resulting in discrepancies between design concepts and realized works. As public appreciation for art deepens, expectations for precision and detail in sculptures have escalated. The advent of 3D laser scanning technology (*Haleem et al., 2022*) introduces a groundbreaking approach to sculpture design.

3D laser scanning technology operates by rapidly emitting and receiving laser light. A diode produces and emits laser pulses, which travel to the environment or object's surface, reflect, and are captured by a receiving lens. The distance is computed using the time difference between emission and reception, alongside the speed of light. The spatial direction of the laser pulse is determined by the instrument's internal angle-measuring prism, allowing for the calculation of the object's 3D coordinates from its distance, horizontal angle, and vertical angle (*Li et al., 2022*). This process facilitates the swift and precise collection of three-dimensional data. Unlike traditional design methods that often necessitate multiple revisions to achieve the desired outcome, the 3D laser scanning-based design process significantly reduces the design cycle and enhances productivity. This innovative technology offers designers fresh perspectives and opportunities to push creative boundaries, encouraging the exploration of varied design styles and forms.

Measuring large or complex sculptures presents significant challenges due to the intricate forms they often possess, such as curved surfaces, indentations, and detailed features. Traditional 3D laser scanning methods may struggle to capture these nuanced variations accurately and may yield noisy or incomplete data sets. The recent development of object segmentation measurement (*Bose, Wang & Grimson, 2007*) addresses the complexities of measuring intricate objects by capturing localized parameters and merging multiple sets of localized data to construct a comprehensive object model.

The core concept of 3D splicing techniques involves transforming the local coordinate systems of various physical parameters and consolidating them into a single data set within a unified coordinate system. A critical algorithm within this domain is the Iterative Closest Point (ICP) algorithm (*Zhang, Yao & Deng, 2021*), a prominent approach in computer vision and computational geometry. ICP is employed to align two 3D point sets by iteratively identifying the optimal rigid transformation (comprising rotation and translation) to minimize a specified metric, such as distance, between the transformed and the reference set. Consequently, ICP has gained extensive application in object model alignment and has spurred the development of numerous enhanced algorithms for more efficient and precise measurement techniques.

The splicing speed of the ICP algorithm significantly declines when dealing with large sculptures, particularly when substantial data is required for testing. This is due to the algorithm's necessity to calculate the nearest point distances between each pair of points and estimate transformation matrices based on these distances. As data volume increases, the number of point pairs requiring computation escalates quadratically, sharply heightening computational complexity. Consequently, the algorithm becomes highly time-consuming when handling extensive point cloud data, potentially failing to meet real-time requirements.

Furthermore, the large volume of data may negatively impact the algorithm's accuracy. Given constraints on computational resources, the algorithm might be unable to incorporate all intricate details when processing vast quantities of data, leading to the potential loss of significant local features during the alignment process and ultimately affecting the final accuracy of the alignment. Additionally, the ICP algorithm's sensitivity to the choice of the initial transformation matrix can result in suboptimal performance. An inaccurate initial alignment may cause the algorithm to converge to a local optimum, producing subpar alignment results. Consequently, the ICP algorithm faces the drawbacks of slow splicing speed and lower accuracy when directly applied to large data sets.

This study evaluates the performance of the Fragment Measurement Iterative Closest Point Feature Point Matching (FM-ICP-FPM) algorithm in three-dimensional scanning and registration tasks, with a particular focus on feature point matching in sculpture design. Currently, feature point matching in sculpture design faces numerous challenges, such as the difficulty in accurately extracting feature points due to complex shapes and the impact of lighting variations on the stability of feature points. These issues limit the applicability of traditional algorithms in sculpture design. However, the FM-ICP-FPM algorithm significantly improves registration accuracy by optimizing the feature point matching process, providing more accurate three-dimensional models for sculpture design. The article presents the following contributions:

(1) Development of a feature point rough splicing algorithm: The algorithm achieves efficient rough splicing by attaching feature points to the overlapping regions of the two viewpoints of the sculpture under examination. Subsequently, a 3D measurement system separately measures the two local surfaces to acquire the physical point cloud data of the two surfaces.

(2) Utilization of intra-block gradient histograms: After segmenting the range near the feature points, the algorithm constructs feature vectors using intra-block gradient histograms, enhancing the scale and rotational stability characteristics of the point cloud data acquired through 3D laser scanning.

(3) Creation of an exact feature point matching algorithm: Building on rough splicing, the algorithm selects precise matching point pairs from the initial rough feature point pairs based on initial feature point matching and distance threshold constraints. The process adds points of the accurate matching region's range to the set of exact matching point pairs in the two surface point clouds.

In this article, the data acquisition and processing in three-position laser scanning technology and the current status of the corresponding engineering applications are presented in "Related Works". The FM-ICP-FPM algorithm constructed in this article is introduced in "Model Design". "Real-Time Image Matching" describes the experimental results and discusses the performance of the FM-ICP-FPM algorithm, compares it with existing feature point matching algorithms, and analyzes the time-consuming performance of the feature point matching algorithm and its impact on the correction system. "Experiments and Analysis" concludes with a discussion of the performance of the improved feature point matching algorithm constructed in this article and the implications for the construction of correction systems in sculpture design.

## RELATED WORKS

### 3D laser scanning data

For effective model design, 3D laser scanning technology requires accurate acquisition and processing of point cloud data. Raw data from 3D laser scanners often entail large volumes, high residuals, unstructured and uneven sampling distributions, and incompleteness. Post-processing is essential before the data can be used in design applications. Key advancements in the intelligent processing of point cloud data encompass point cloud data model integration (*Tian et al., 2021*), data quality enhancement, data characterization and extraction (*Chen et al., 2024*), and data model reconstruction (*Chen et al., 2018*).

*Wang et al. (2022)* explores external data acquisition methods of terrestrial 3D laser scanning technology from an engineering perspective, summarizing the advantages, disadvantages, and corresponding scopes of application for each technique. *Kang et al. (2023)* discusses point cloud data acquisition and processing technologies for the Nanjing subway tunnel project, yielding high-quality point cloud data. *Benchekroun (2022)* applies 3D laser scanning to cultural heritage sites, examining heritage buildings' rapid laser scanning method. It rapidly scans the National Southwest United University site, offering a reference for archiving and digitizing historical relics and sites.

Despite its speed and sampling density advantages over traditional methods, 3D laser scanning produces point cloud data with high redundancy, nonlinear error distribution, and incompleteness, challenging intelligent processing. *Wang, Yin & Jing (2023)* investigates standard point cloud filtering algorithms based on the PCL library to address these issues, analyzing algorithm parameter settings for optimal filtering effects. It identifies the best waveform algorithm and proposes a hybrid approach combining Gaussian filtering.

For complex building point cloud data segmentation challenges such as low efficiency, poor noise resistance, and low accuracy, *Xu, Li & Chen (2022)* focuses on building roof structures. The method improves efficiency by collecting and comparing accurate building point cloud data.

### Feature point matching algorithm based on 3D laser scanning technology

The growing advancement of 3D laser scanning technology has led to an expansion of its application areas. *Mi et al. (2023)* reports the application of this technology in obtaining substantial point cloud data of pit enclosure walls. Through meticulous point cloud processing and modeling, the study conducted in-depth deformation analysis using advanced software. The technology excelled in matching feature points and accurately revealing three-dimensional overall deformation and two-dimensional local deformation of the pit enclosure wall. Additionally, the study compared the differences between the two monitoring techniques regarding deformation magnitude and monitoring accuracy, providing critical data support for engineering practice. However, *McDonald, Robinson & Tian (2022)* reveals that while 3D scanning demonstrates clear benefits in scope and

efficiency for pit monitoring, it lags behind traditional monitoring methods regarding accuracy and interference level, which introduces potential bias in monitoring outcomes.

Integrating 3D laser scanning technology and building information modeling technology has emerged as a significant research focus in the architectural field. Scholars are exploring novel intelligent construction quality inspection methods and assessment techniques for more precise corrective system design. *Yue et al. (2024)* employs 3D laser scanning technology to match feature points for construction quality inspection. *Xu et al. (2023)* explores terrestrial 3D laser scanners in building stereoscopy, collecting 3D spatial data of target building surfaces. The study suggests extracting the building's center axis from the point cloud using a random sample consensus algorithm to fit straight lines and an overall least squares method to fit planes, achieving precise feature point matching for building flatness and perpendicularity.

*Wang et al. (2021)* introduces a comprehensive assessment of wall flatness through 3D laser scanning. By calculating the curvature of wall contour lines and analyzing the relationship between curvature features and flatness, efficient feature point matching of the wall point cloud was achieved using the least squares method. Moreover, *Yao et al. (2022)* utilizes SIFT operator (*Pan et al., 2024*) and ORB operator (*Yang et al., 2019*) to extract image features from infrared and visible images for image matching, although large rotation angles between images may reduce matching accuracy. *Huang et al. (2023)* suggests a stereo-matching method based on bootstrap image and adaptive support domain, offering less operational difficulty but lower resistance to varying illumination conditions. *Wei & Meng (2023)*, *Wu & Chiu (2023)*, *Guo et al. (2024)* proposes a pattern of matching light and color information features to achieve image matching with better effects; however, the matching effect may need validation for pictures with poor quality.

Despite the advancements in feature point matching methods, specific challenges persist, particularly in specialized application scenarios such as sculpture design. Sculptures often exhibit complex and variable shapes, making direct measurement nearly unfeasible. In response to this issue, this article introduces an innovative nearest point iteration algorithm grounded in the principle of feature point matching. The algorithm aims to enhance efficiency and accuracy in feature point matching, offering robust technical support for 3D modeling and shape correction analysis within sculpture design.

## MODEL DESIGN

The FM-ICP-FPM algorithm introduced in this article is outlined in Fig. 1. Its core principle revolves around the precise measurement of two localized surfaces of the sculpture using a 3D measurement system. This allows the algorithm to utilize object segmentation measurement techniques to extract comprehensive point cloud data, enabling the initial stitching of the feature points.

Following this, the algorithm segments the area surrounding the feature points in detail and computes the histogram of gradients within each segment to construct precise feature vectors. These feature vectors are then employed to match point pairs with the initial feature points, filtering the exact feature point matching pairs from the initial feature point

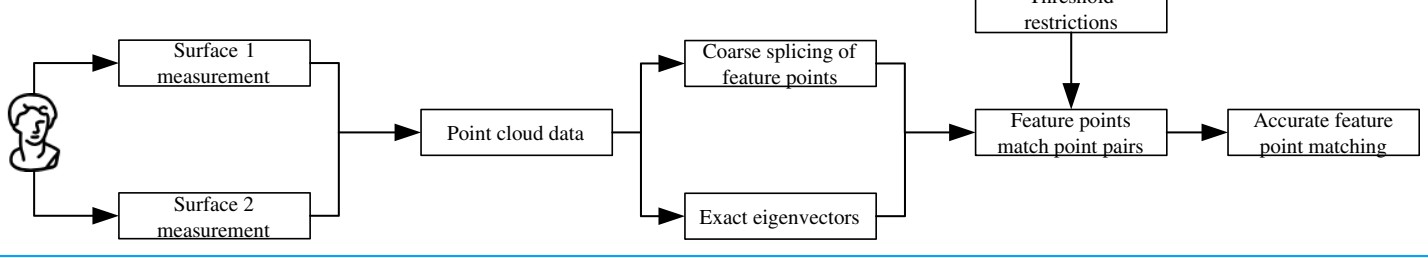

**Figure 1  Framework of FM-ICP-FPM algorithm.**               

pairs based on rigorous distance threshold constraints. This approach guarantees precise and efficient feature point matching, supporting 3D modeling and shape correction analysis in sculpture design.

## Feature point pairs

In 3D laser scanning technology, feature points are crucial identification markers in 3D stitching. In this article, feature points are designated as the overlapping region between two viewpoints of the sculpture under examination. Using a three-dimensional laser scanning system, two local surfaces of the sculpture are scanned to obtain physical point cloud data for both surfaces.

To ensure the accuracy of these measurements, the feature point sets of the two distinct local surfaces are pre-established. These sets are defined as follows:

$$M = \{m_i\}, i = q, 2, ..., k \tag{1}$$
$$N = \{n_i\}, i = 1, 2, ..., k. \tag{2}$$

Then, according to the spatial feature invariance of the feature points, n sets of matching point pairs $U = \{u_i, u_i \in M\}$ and $V = \{v_i, v_i \in N\}$ can be obtained. $U$ and $V$ are the subsets of the 3D data point sets obtained under the two surfaces, respectively, and $u_i$ and $v_i$ are one-to-one correspondences. On this basis, according to the principle of 3D data stitching technique, let the rotation matrix R and translation vector T satisfy the following objective function to obtain the minimum value:

$$e^2(R, T) = \frac{1}{n} \sum_{i=1}^{n} ||v_i - (R \times u_i + T)||^2. \tag{3}$$

To obtain the coordinate transformation matrix, according to the feature point sets $X = \{x_i\}, i = 1, 2, ..., n$ and $Y = \{y_i\}, i = 1, 2, ..., n$ in two views of 3D space, the rotation matrix R and translation matrix can be obtained by the following Singular Value Decomposition (SVD) decomposition method, which ensures that the splicing is free from aberrations and has relatively high accuracy. Thus, Eq. (3) can be varied under the SVD decomposition method as:

$$e^2(R, T) = \frac{1}{n} ||Y - R \times X - T \times H^T||^2 \tag{4}$$

where $H = \{1, 1, ..., 1\}^T$, and according to the regular array (Eq. (5)):

$$K = I - \frac{1}{n} H \times H^T \tag{5}$$

SVD analysis can be performed using $K^2 = K^T = K$ and $P = \frac{1}{n} Y \times K \times X^T$ to obtain R. For spatial point sets, the calculation is done separately:

$$u_x = \frac{1}{n} \sum_{i=1}^{n} x_i \tag{6}$$

$$u_y = \frac{1}{n} \sum_{i=1}^{n} y_i. \tag{7}$$

Calculate the translation vector $T = u_y - R \times u_x$. After obtaining R and T, any point $y_i$ in the feature point set Y can be transformed into X by the following equation:

$$y_i = x_i \times R + T. \tag{8}$$

The two data sets are eventually transformed to the same coordinate system, resulting in a coarse splice.

## REAL-TIME IMAGE MATCHING

The current method computes binary features to effectively match feature points in 3D laser scanning data, which is inefficient and prone to errors. This article proposes a technique ensuring scale and rotation invariance in feature points extracted from 3D laser point clouds.

The approach involves segmenting the area around the feature points into smaller chunks and operating a gradient histogram within each chunk to create feature vectors, as illustrated in Fig. 2. Firstly, the area near the feature points is segmented, with the size of each segment adjustable according to the complexity of the sculpture and the distribution of feature points. Then, within each segment, each pixel's gradient direction and magnitude are calculated, and the number of pixels in different directions is counted to obtain a gradient histogram. A feature vector is constructed based on each segment's gradient histogram. Following this, a binary feature matching scheme based on fast nearest neighbor retrieval is employed to eliminate incorrectly matched feature points using a model of random sampling consistency. The method for performing nearest-neighbor distance operations is as follows:

$$L(U_a, U_b) = \sum_{j=0}^{511} \varpi_j \oplus \alpha_j \tag{9}$$

The j-th match in the binary feature description $U_a$ of a random extreme point in the reference 3D laser scanning image is $\varpi_j$, and the j-th match in the binary feature description $U_b$ of a random extreme point in the image to be matched is $\alpha_j$.
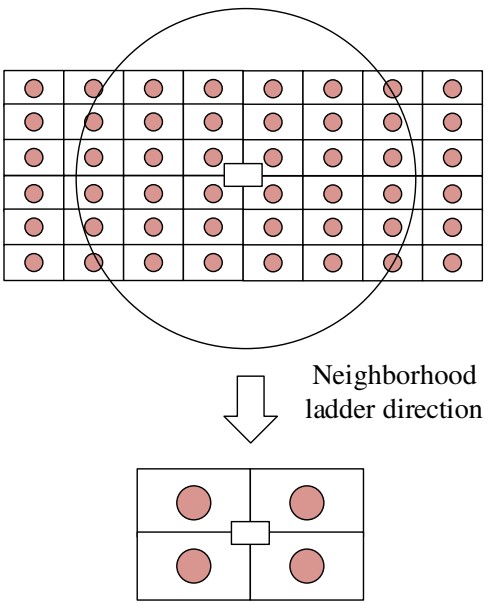

Neighborhood
ladder direction

Core point matrix vector

**Figure 2 Core point matrix vector.**

Given the high-dimensional nature of 3D laser scanning image features, employ a scalable nearest neighbor algorithm to extract the coarse matching point pairs of image features. Subsequently, a consistent random sampling algorithm was used to extract the feature-matching point pairs, ensuring a robust and precise match. Finally, facilitate the fast matching of all feature matching point pairs by eliminating erroneously matched feature points, thereby achieving accurate image matching. The specific steps involved in this process are as follows:

(1) Retrieve the nearest neighbor and second nearest neighbor of the feature point U in the image. Set the nearest neighbor and second nearest neighbor to be the feature point $c_1, c_2$ in the image to be matched, and the Hamming distance between it and the feature point U is $H_{U,c_1}$ and $H_{U,c_2}$ in order.

(2) Match the feature point pairs and extract the matching point pair set. After matching the set of matched point pairs, the matching ratio is:

$$H = \frac{H_{U,c_1}}{H_{U,c_2}} \tag{10}$$

If H is not greater than 0.55, feature point matching is successful.

(3) Remove unsuccessful matching feature points to realize fast real-time 3D laser scanning image matching.

## Precise splicing and iterative matching

Following the initial rough splicing and rapid real-time matching of 3D laser scanning images, the process leverages the initial matching point pairs derived from feature points to establish the splicing position of the two local surfaces as the starting point for localization.

The specific process is shown in Fig. 3. To achieve iterative matching for precise stitching, enhancements are made to the existing ICP algorithm. The rough feature point pairs are filtered based on a distance threshold, ensuring that only points within the correct matching region are included in the exact matching point pairs.

Although conventional ICP algorithms may suffer from slow computation speeds, this limitation is addressed by setting a specific neighborhood dimension as the distance threshold and calculating distances between any two points within that range. This approach significantly reduces splicing time and accelerates computation by processing only point cloud data within the neighborhood that falls within the small distance threshold.

Therefore, in this article, we propose to use the mean square distance between two points as the metric of the distance between nearest neighbors and let $X^{m-1} = \{x_i\}, i = 1, 2, \ldots, r$ and $Y^{m-1} = \{y_i\}, i = 1, 2, \ldots, r$ be the data point sets of the nearest points obtained in m-1 iterations, respectively. Then the mean square distance between the point sets $X^{m-1}$ and $Y^{m-1}$ is:

$$S = \frac{1}{r} \sum_{i=1}^{r} ||x_i - y_i||^2 \tag{11}$$

where $r$ is the number of point cloud data in the overlapping surfaces, and the convergence speed of the algorithm can be significantly accelerated by using this nearest point distance metric. Therefore, the error threshold S is set in the iterative matching, and the coordinate transformation matrices $R_0$ and $T_0$ the 3D data splicing are solved. The coordinate transformations of the two groups of point cloud data are carried out according to the initial coordinate transformation relationship. Finally, the distance between the two groups of point cloud data after the transformations is obtained $S_1$. According to the distance threshold constraints, the closest matching point pairs in the two groups of matching point clouds are searched. After eliminating the invalid matching pairs, the closest matching pairs are then optimized using the four-element algorithm with correction coefficients to solve for the transformation parameters $R_1$ and $T_1$. The above two sets of point cloud data are transformed again using $R_1$ and $T_1$, the distance between them after this transformation is found out at $S_2$. If $S_2$ is less than the initial threshold S, then stop searching. Otherwise, the search continues until $S_i$ is smaller than the initial threshold. Eventually, the most accurate feature point match is found when the search stops.

## EXPERIMENTS AND ANALYSIS

In this section, the performance of the proposed FM-ICP-FPM algorithm is meticulously analyzed, focusing on assessing its efficiency in feature point matching and its overall impact on the correction system. This is achieved through a comparative study with existing feature point matching algorithms found in the literature, references explicitly (*Wang et al., 2021*; *Yao et al., 2022*), and *Wei & Meng (2023)*. For clarity, the algorithm from *Wang et al. (2021)* that employs the least squares method for feature point matching is designated as Least Squares Feature Point Matching (LS-FPM). The feature point matching algorithm combining Scale-Invariant Feature Transform (SIFT) and Oriented
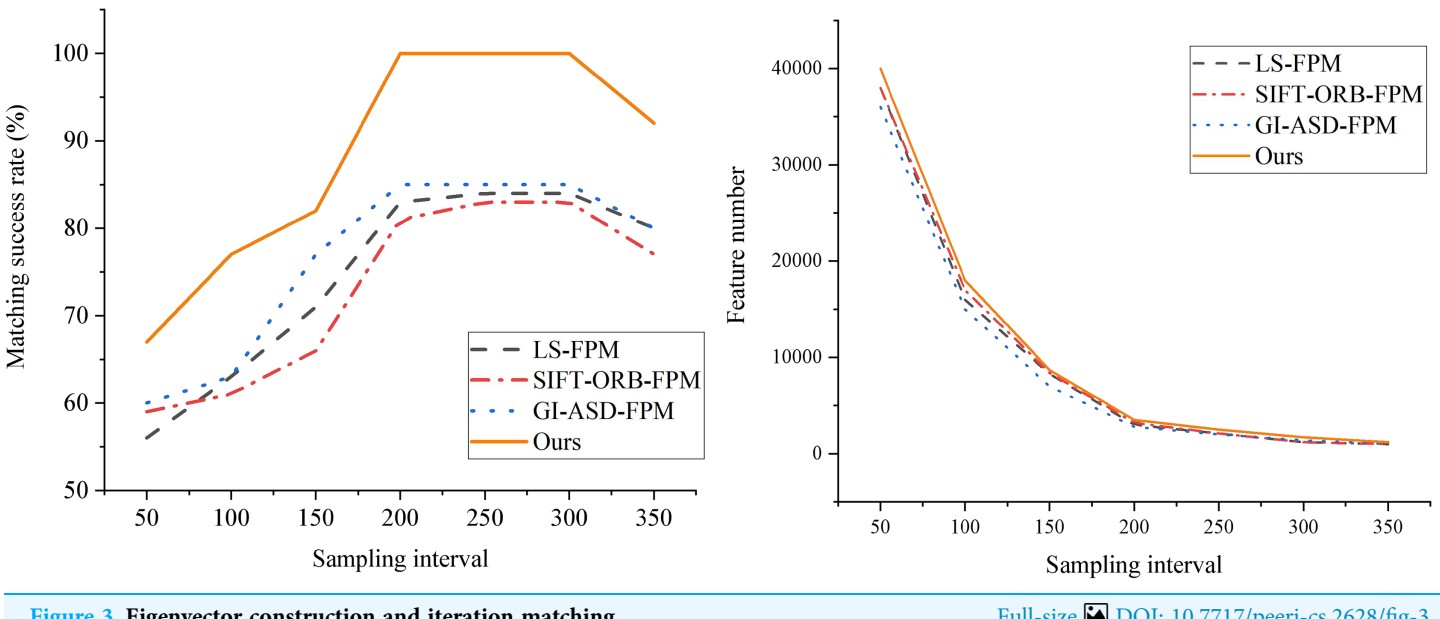

**Figure 3 Eigenvector construction and iteration matching.**

FAST and Rotated BRIE (ORB) operators from *Yao et al. (2022)* is called Scale-Invariant Feature Transform and Oriented FAST Rotated BRIEF Feature Point Matching (SIFT-ORB-FPM). Meanwhile, the feature point matching method utilizing guided images and adaptive support domains from *Wei & Meng (2023)* is denoted as Guided Images Adaptive Support Domain Feature Point Matching (GI-ASD-FPM).

## Evaluation indicators

To assess the accuracy of alignment, this study employs two metrics: mean rotation angle error (MRAE) and mean translation error (MTE). MRAE is utilized to quantify the degree of deviation in the rotation angle during the registration process. This error value is obtained by calculating the rotational angle errors at multiple measurement points and then averaging them. MTE, on the other hand, describes the translational discrepancy between the measurement results and the reference values. This type of error can be expressed as the average of the translational differences between all measurement and reference values. In evaluating registration accuracy, the mean translation error provides insights into the positional offsets generated during registration. Generally, a smaller mean translation error indicates more precise registration results, while a larger error may suggest significant deviations in the registration process. Assuming there are a total of n measurements, the formula for calculating MRAE is as follows:

$$\text{MRAE} = \frac{1}{n}\sum_{i}^{n} \frac{|R_i - T_i|}{T_i} \tag{12}$$

where $R_i$ is the actual output angle and $T_i$ is the theoretical output angle.

The formula for MTE is as follows:

$$MTE = \frac{1}{n}\sum_{i=1}^{n}|M_i - Value| \qquad (13)$$

where $M_i$ is the actual measured value and value is the reference value.

## Comparative analysis

By examining the overlapping region of two views of a sculpture, an advanced 3D measurement system is employed to capture the two local surfaces precisely, enabling the alignment and comparison of feature points. Successful feature alignment offers high-quality initial values for the subsequent ICP algorithm throughout the alignment process, playing a pivotal role in enhancing overall alignment accuracy. Given the influence of the sampling interval on the volume of point cloud data, tests are conducted at various intervals to assess the matching success rates of different algorithms and the number of feature points. The results displayed in Fig. 4 are obtained through thorough comparisons and experimental analyses.

The figure illustrates a strong correlation between the matching success rate and the sampling interval. When the distance threshold is relatively tiny, the matching accuracy of both algorithms is low. This is because a minimal threshold limits the number of matching points. As the distance threshold increases, the matching accuracy of the FM-ICP-FPM algorithm rapidly improves and reaches an optimum within a specific threshold range. In contrast, the traditional ICP algorithm exhibits slower improvement in matching accuracy when the distance threshold is significant, and it is prone to being affected by incorrect matching points. Across all tested distance thresholds, the matching accuracy of the FM-ICP-FPM algorithm is higher than that of the traditional ICP algorithm. By introducing feature point matching and gradient histogram feature vectors, the FM-ICP-FPM algorithm enhances the accuracy and robustness of matching. Especially when the distance threshold is moderate, this algorithm can screen out more precise matching points, thereby improving the overall matching accuracy. Notably, the FM-ICP-FPM algorithm proposed in this article demonstrates superior performance at sampling intervals of 200 and 300, achieving a success rate of 100%—considerably higher than the 83% peak success rate of the SIFT-ORB-FPM algorithm. This finding indicates that the FM-ICP-FPM algorithm boasts exceptional accuracy and stability in feature point matching and alignment, offering a promising and practical approach to 3D reconstruction and alignment.

After a comprehensive analysis of the impact of sampling intervals on the success rate of feature point matching, this section proceeds to select several representative sampling intervals and conduct comparative tests on the MRAE and MTE performance metrics across different algorithms. As depicted in Fig. 5, the proposed FM-ICP-FPM algorithm demonstrates clear superiority across all metrics. It is evident from the figure that the performance of LS-FPM, SIFT-ORB-FPM, and GI-ASD-FPM in terms of MRAE and MTE is notably inferior to that of the FM-ICP-FPM algorithm.

A notable observation emerges from the data: as the number of sampling points increases from 200 to 300, the MRAE and MTE performance metrics of the compared

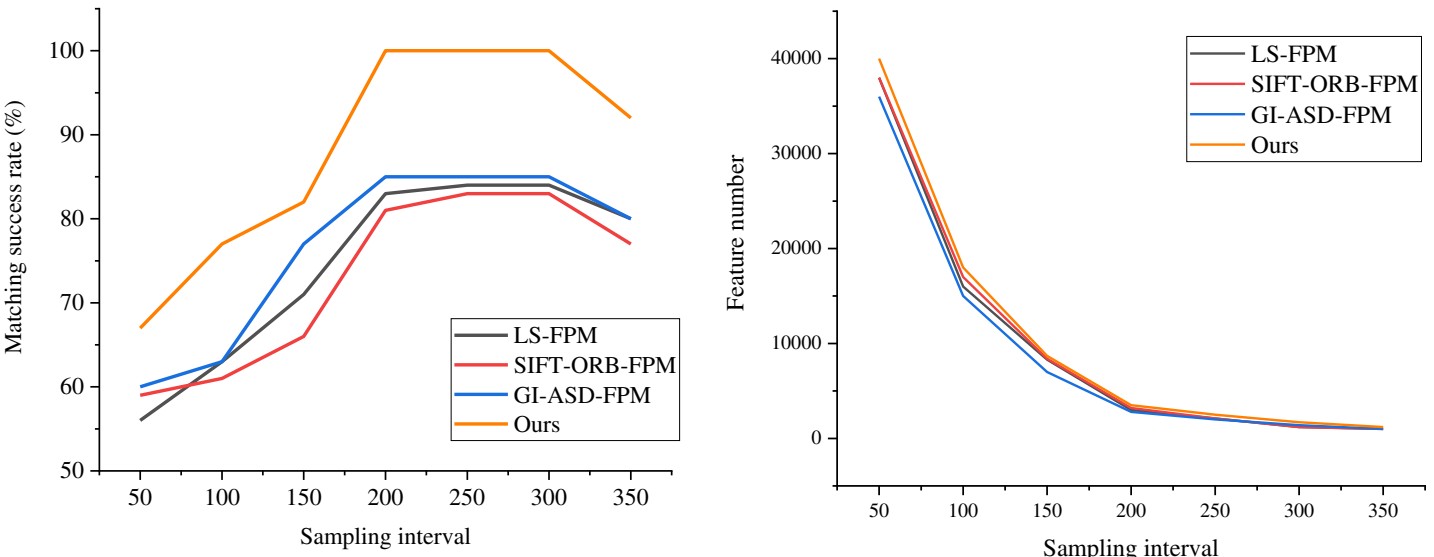

**Figure 4 Comparison of the matching success rate and feature points of each algorithm at different sampling intervals.**

algorithms undergo a marked decline. Specifically, while the GI-ASD-FPM algorithm demonstrates improvements relative to LS-FPM and SIFT-ORB-FPM, its performance still sees a significant drop, with the MTE rising from 201 at a sampling interval of 200 to 479 at an interval of 300. In stark contrast, the FM-ICP-FPM algorithm stands out, achieving a remarkable matching success rate of 100% at a sampling interval of 200. Moreover, its MTE of 154 and MRAE of 0.065 are considerably lower than those of the other algorithms, highlighting its superior performance in terms of accuracy and reliability.

This result demonstrates that the FM-ICP-FPM algorithm effectively minimizes alignment error and enhances alignment accuracy while maintaining a high matching success rate. These findings offer a more reliable and efficient approach to feature point matching and correction system design in sculpture creation.

### Feature point matching efficiency test

Figure 5 provides an in-depth look at the performance of the proposed FM-ICP-FPM algorithm at a sampling interval of 200, illustrating its accuracy before and after ICP alignment integration. The figure shows that the alignment accuracy significantly improves after performing two-by-two ICP fine alignment on four pairs of point clouds and optimizing the ICP algorithm.

Initially, the feature-matching alignment stage achieves a certain degree of alignment, yet its accuracy remains constrained. However, by merging the initial alignment results from feature matching with the standard ICP algorithm, more precise alignment results are achieved. This is due to the iterative optimization of the transformation matrix by the ICP algorithm, which leads to a more accurate point cloud alignment.

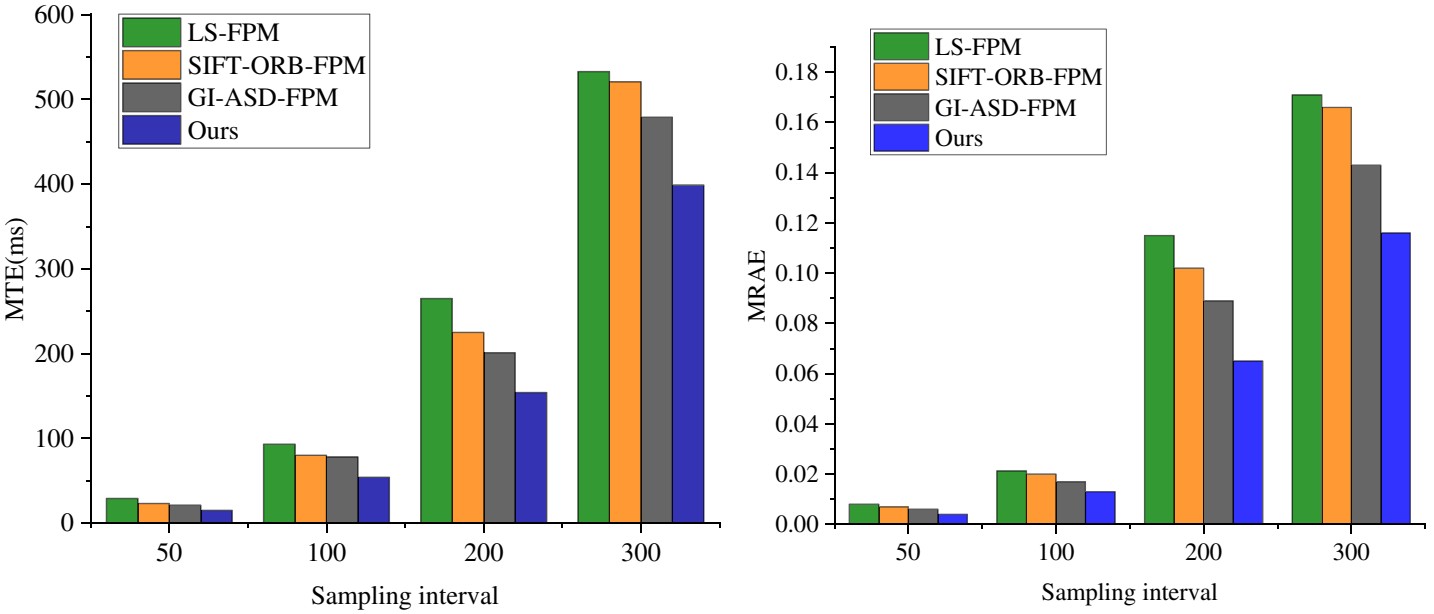

**Figure 5 Comparison of MRAE and MTE of each algorithm at different sampling intervals algorithm.**

Notably, the FM-ICP-FPM algorithm delivers initial solid values for the subsequent ICP algorithm during the feature matching stage, facilitating the ICP algorithm's rapid convergence to the optimal solution and further enhancing alignment accuracy. As a result, the combination of the feature matching and ICP algorithms enables the proposed FM-ICP-FPM algorithm to achieve high-precision 3D point cloud alignment without compromising alignment speed.

This section presents two experiments to thoroughly assess the FM-ICP-FPM algorithm's operational efficiency in handling the 3D laser scanning image matching task. In Experiment E1, the number of core feature points of 3D laser scanning images to be matched is equivalent to the number of matching point pairs. In Experiment E2, the number of core feature points of 3D laser scanning images to be matched is double that of the matching point pairs.

Under the conditions of Experiment E1, as depicted in Fig. 6A, it is evident that the elapsed time required for image matching increases proportionally as the number of core feature points of the 3D laser scanning image to be matched gradually rises from 10 to 70. Notably, when the number of core feature points and matching point pairs reaches 70, the FM-ICP-FPM algorithm's matching time is a mere 423 ms, with an MTE of 201 mm.

Conversely, in Experiment E2, illustrated in Fig. 7B, as the number of core feature points increases from 40 to 50 and the corresponding matching point pairs rise from 20 to 25, the matching elapsed time sees a noticeable jump from 204 to 256 ms. At the same time, the MTE climbs from 121 mm to 152 mm. This increase in matching elapsed time is attributed to the substantial increase in the number of matching point pairs, which poses challenges in memory and cache management.

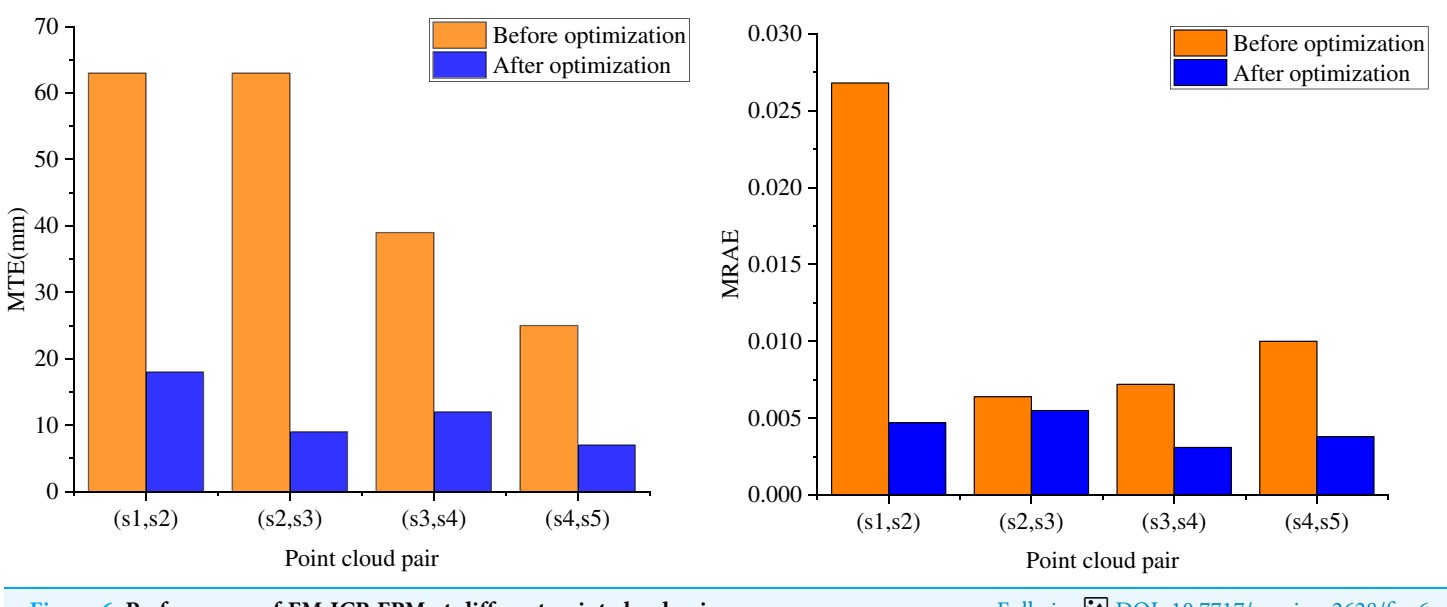

**Figure 6 Performance of FM-ICP-FPM at different point cloud pairs.**

Figure 7C demonstrates the results of the MRAE metrics for both experimental setups. It is evident that as the number of matching points increases, the MRAE also rises, although the MRAE in E1 consistently remains lower than that in E2. Despite the increasing matching points, the overall time to address the 3D laser scanning image matching problem remains efficient.

## DISCUSSION

According to the experimental analysis in "Comparative Analysis" and "Feature Point Matching Efficiency Test", the FM-ICP-FPM algorithm achieves a 100% matching success rate, with an average translation error of only 154 mm and an average rotation angle error of 0.065. Such high-precision feature point matching significantly enhances the reconstruction accuracy and efficiency during the three-dimensional reconstruction of sculptures.

Improved feature point matching performance greatly benefits the accuracy of 3D reconstruction. In the context of sculpture reconstruction, precise feature point matching is essential to maintain the authenticity and detail of the model. The FM-ICP-FPM algorithm enables accurate feature point matching on the sculpture's surface, allowing the system to capture the sculpture's shape and intricate details with greater precision. This is particularly important for applications requiring high-fidelity restoration of the sculpture's original appearance, such as digital preservation of artworks and virtual museum exhibitions.

Moreover, enhanced feature point matching performance optimizes the automation of the sculpture correction system. Precise adjustments to the sculpture's posture and position are often necessary in the production and restoration of sculptures. By swiftly and accurately identifying and matching feature points, the system can perform these

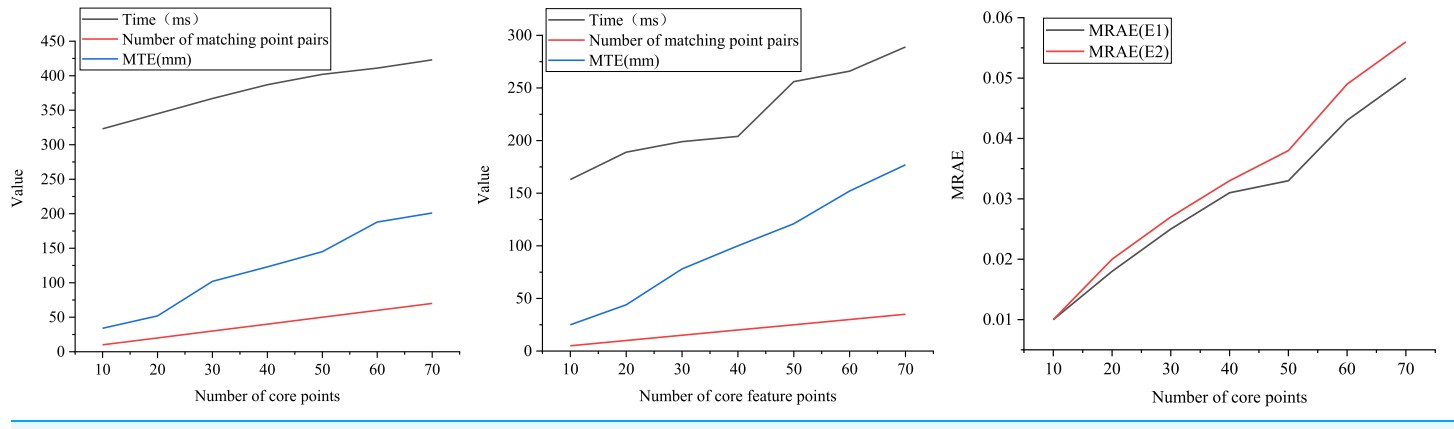

**Figure 7** Matching efficiency of feature points.                             

adjustments more intelligently and automatically, enabling efficient automated correction. This reduces manual intervention and operational challenges while also improving correction accuracy and consistency, leading to a significant boost in work efficiency.

Additionally, enhanced feature point matching performance strengthens the sculpture correction system's adaptability to complex environments and lighting conditions. In real-world applications, sculptures may be exposed to various environmental backgrounds and lighting scenarios, which can challenge feature point recognition and matching. By improving feature point-matching performance, the system can better handle these complexities and achieve precise correction across diverse environmental conditions.

In summary, the outstanding performance of the FM-ICP-FPM algorithm in feature point matching offers robust technical support for the 3D reconstruction and correction systems of sculptures. By continually enhancing feature point matching performance, we can anticipate achieving more realistic and detailed 3D models of sculptures and more intelligent and efficient correction systems in the future. This advancement will infuse new vitality into the digital evolution of the sculpture art field and foster the progress and broader application of related technologies.

## CONCLUSION

This study introduces a swift and precise feature point matching algorithm, FM-ICP-FPM, grounded in object slice measurement and the ICP algorithm. The method involves assigning feature points to the overlapping regions of the two viewpoints of the subject sculpture. By incorporating feature markers on the reconstructed surface, we derive the initial transformation matrix through object slice measurement and preliminary alignment. The 3D laser scanning image's core features are ultimately extracted and aligned, facilitating accurate stitching and feature point matching. Experimental results indicate that the FM-ICP-FPM algorithm significantly enhances the speed of nearest-neighbor point searches, thereby boosting retrieval efficiency and yielding precise stitching outcomes. The approach achieves high splicing accuracy and is apt for complex and sizable objects such as sculptures, showcasing strong convergence capabilities. Sculptures or

objects in dynamic environments, such as positional shifts or shape deformations, may change over time. This can pose challenges for algorithms during the feature point matching and registration stages, as the initial feature point matches and transformation matrices may no longer be applicable. Additionally, other objects or obstructions may be present in dynamic environments, interfering with the accuracy and completeness of three-dimensional laser scanning. Furthermore, factors such as lighting variations and surface materials can also affect the extraction and matching of feature points. In the future, we will introduce more robust feature point extraction methods, adaptive parameter adjustment mechanisms, and registration strategies for dynamic environments to improve our solutions.

### Funding
This work is funded by Special Project of "Serving Rural Revitalization" in Guangdong Province in 2019 (Humanities and Social Sciences), project number: 2019KZDZX2027. The funders had no role in study design, data collection and analysis, decision to publish, or preparation of the manuscript.

### Grant Disclosures
The following grant information was disclosed by the authors:
Special Project of "Serving Rural Revitalization" in Guangdong Province in 2019 (Humanities and Social Sciences): 2019KZDZX2027.

### Competing Interests
The authors declare that they have no competing interests.

### Author Contributions
- Xiaoxiong Zheng conceived and designed the experiments, analyzed the data, prepared figures and/or tables, and approved the final draft.
- Zhenwei Weng performed the experiments, performed the computation work, authored or reviewed drafts of the article, and approved the final draft.

### Data Availability
The code is available in the Supplemental File. The data is available at Zenodo: varldskulturmuseerna. (2018). Wooden Sculpture [Data set]. Zenodo. https://doi.org/10.5281/zenodo.10330814.

### Supplemental Information
Supplemental information for this article can be found online at http://dx.doi.org/10.7717/peerj-cs.2628#supplemental-information.

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
