# Peer review of "Design of an enhanced feature point matching algorithm utilizing 3D laser scanning technology for sculpture design"

_PeerJ Computer Science, doi:10.7717/peerj-cs.2628_

## Round 0.1 · original submission · Major Revisions

Dear Author,

Thank you for submitting your manuscript on the FM-ICP-FPM algorithm for feature point matching in sculpture design. The integration of 3D laser scanning and the proposed algorithm presents a compelling advancement for precision and automation in art restoration and design. However, the technical experts are of the view that your manuscript needs a couple of major improvements as outlined, Please consider them carefully and my suggestions are below which will help you to polish your article.

Please add a discussion on the potential versatility of the FM-ICP-FPM algorithm
please add a comparative analysis with other widely used feature point matching methods.

Clarify the level of automation involved in the algorithm and whether manual adjustments are necessary during alignment and feature matching

Reviewer 1 ·

Basic reporting

The manuscript introduces an enhanced feature point matching algorithm for sculpture design using 3D laser scanning, which is a pertinent topic in the context of computer vision and digital art. The FM-ICP-FPM algorithm achieves a high matching success rate, and the experimental results are promising. However, there are several key areas where the paper could be strengthened to better support its claims.
(1) Although the paper describes the basic steps of the FM-ICP-FPM algorithm, it still lacks details, especially in the neighborhood dimension setting of the gradient histogram generation and iterative matching stage. Adding pseudocode or a more detailed description of the algorithm flow would help to improve the reproducibility of the study.
(2) The experiments are mainly conducted on static sculptures, while the 3D scanning environment is usually dynamically changed in practical applications. It is recommended to add validation on dynamic 3D laser scanning data or under different lighting conditions to enhance the generality of the algorithm.

Experimental design

(3) The paper does not explicitly discuss the potential limitations of the proposed algorithm. For example, the dependence of the algorithm on parameter Settings, such as distance thresholds and histogram split sizes, may affect its applicability in different scenarios, and this aspect should be further analyzed.
(4) More detailed algorithm flowcharts or pseudocodes are provided, especially in the feature vector construction and iterative matching sections, so that readers can better understand the implementation details of the algorithm.
(5) Add more descriptive captions to Figures 3 and 4, summarizing key findings and explaining what trends or differences are being shown.

Validity of the findings

(6) It is recommended to test the algorithm in dynamic scenes or under different lighting conditions to verify its robustness in complex environments.
(7) Limitations of the algorithm, such as sensitivity to specific parameters, are explicitly discussed, and possible future research directions, such as adaptive parameter tuning or more efficient iterative matching strategies, are suggested.
(8) The language of some paragraphs can be more concise to improve the readability of the paper. In addition, it is recommended to describe technical details in more formal terms.

Additional comments

I recommend the manuscript be revised before publication to address the identified issues.

Reviewer 2 ·

Basic reporting

The work introduces innovative aspects in terms of matching efficiency, especially for complex sculptural shapes. The study is well-motivated, and the problem is clearly defined, with a structured approach to address the challenges of 3D point cloud alignment and matching. However, some areas need improvement to enhance the manuscript's overall quality.
-The abstract is informative but could benefit from more clarity in conveying the significance of the results. Consider providing a brief explanation of terms such as "sampling interval" and why a 100% matching success rate is impressive. Clarifying the implications of the reported Mean Translation Error (MTE) and Mean Rotation Angle Error (MRAE) values would also help make the abstract more accessible.
-The introduction provides good context but could be enhanced by adding a discussion on current limitations or challenges in feature point matching in sculpture design. This would help establish a stronger justification for the proposed FM-ICP-FPM algorithm and highlight the novelty of the contribution.

Experimental design

- The explanation of the FM-ICP-FPM algorithm could be more detailed. Including a pseudocode or step-by-step breakdown of the algorithm would help readers understand its flow and implementation. Additionally, it would be helpful to clarify how the FM-ICP-FPM approach improves upon traditional ICP algorithms, with specific examples.

- In the comparative analysis, the discussion focuses mainly on the performance of the FM-ICP-FPM algorithm but lacks depth in explaining why the other methods perform less effectively. Adding a detailed discussion about the strengths and weaknesses of the baseline algorithms (LS-FPM, SIFT-ORB-FPM, GI-ASD-FPM) compared to FM-ICP-FPM would strengthen the results section.
- The current figures provide useful information, but some could benefit from more descriptive captions that explain the trends and findings depicted. For instance, Figures 3 and 4 could include captions summarizing the key observations. Adding visual comparisons of actual feature point matching results (e.g., images showing matched points before and after applying FM-ICP-FPM) could further illustrate the improvements.

Validity of the findings

-The discussion section could be expanded to address potential limitations of the FM-ICP-FPM algorithm in real-world scenarios, such as the impact of varying environmental conditions or lighting. Additionally, consider discussing potential applications beyond sculpture design, such as in cultural heritage preservation or industrial inspection.

-While MRAE and MTE are defined, a more in-depth explanation of how they are calculated and interpreted would benefit readers who may not be familiar with these metrics. Providing context on why these specific metrics were chosen for evaluation and discussing their importance in feature point matching would add clarity.

-The data availability section mentions that the dataset is accessible. Consider providing more details about how the dataset can be obtained or shared (e.g., a link or repository). Additionally, including code or supplementary materials for reproducing the experiments would enhance the manuscript's reproducibility.

---

## Round 0.2 · Minor Revisions

Dear authors,

Thank you for your re-submission after updating the article. However, some of the changes/amendments need your attention as pointed out by reviewer 1, especially the language improvement. Therefore, we advise you to update the paper and improve the langue then re-submit carefully.

thank you

Reviewer 1 ·

Basic reporting

1- Enhance the introduction section with relevant visuals or diagrams to make it more engaging and visually appealing.
2- Ensure that Figures 1 and 3 are formatted to align with the publication standards, including proper labeling and resolution.
3- Conduct a thorough review of the paper for grammatical errors, sentence structure, and overall language quality to maintain a professional tone.
4- Verify the citation and referencing format to ensure consistency with the required guidelines.

Experimental design

Changes are done

Validity of the findings

Changes are done

Additional comments

Changes are done

Reviewer 2 ·

Basic reporting

The paper seems to be in better overall reporting

Experimental design

The author has done reasonable updates and it looks in an acceptable form

Validity of the findings

overall it can be accepted after the update

---

## Round 0.3 · accepted · Accept

Thank you for revision and updating the Paper in light of the previous comments. I'm pleased to inform you that your manuscript is now being recommended for publication.

Thank you for your contribution

Reviewer 1 ·

Basic reporting

All changes have been completed.

Experimental design

All changes have been completed.

Validity of the findings

All changes have been completed.

Additional comments

All changes have been completed.